# Resveratrol’s Impact on the Chondrogenic Reagents’ Effects in Cell Sheet Cultures of Wharton’s Jelly-Derived MSCs

**DOI:** 10.3390/cells12242845

**Published:** 2023-12-15

**Authors:** Anastasiia D. Kurenkova, Viktoria S. Presniakova, Zlata A. Mosina, Pavel D. Kibirskiy, Irina A. Romanova, Gilyana K. Tugaeva, Nastasia V. Kosheleva, Kirill S. Vinogradov, Sergei V. Kostjuk, Svetlana L. Kotova, Yury A. Rochev, Ekaterina V. Medvedeva, Peter S. Timashev

**Affiliations:** 1Institute for Regenerative Medicine, Sechenov First Moscow State Medical University (Sechenov University), 8-2 Trubetskaya St., Moscow 119991, Russia; 2FSBSI “Institute of General Pathology and Pathophysiology”, Baltiyskaya St. 8, Moscow 125315, Russia; 3Department of Chemistry, Belarussian State University, 14 Leningradskaya St., 220006 Minsk, Belarus; 4Research Institute for Physical Chemical Problems of the Belarusian State University, 14 Leningradskaya St., 220006 Minsk, Belarus; 5Center for Research in Medical Devices (CÚRAM), National University of Ireland Galway, H91 W2TY Galway, Ireland

**Keywords:** Wharton’s jelly mesenchymal stem cells, joint cartilage defect, Y27632, LiCl, resveratrol, P(NIPAM-co-NtBA)-based matrices

## Abstract

Human Wharton’s jelly mesenchymal stem cells (hWJ-MSCs) are of great interest in tissue engineering. We obtained hWJ-MSCs from four patients, and then we stimulated their chondrogenic phenotype formation in vitro by adding resveratrol (during cell expansion) and a canonical Wnt pathway activator, LiCl, as well as a Rho-associated protein kinase inhibitor, Y27632 (during differentiation). The effects of the added reagents on the formation of hWJ-MSC sheets destined to repair osteochondral injuries were investigated. Three-dimensional hWJ-MSC sheets grown on P(NIPAM-co-NtBA)-based matrices were characterized in vitro and in vivo. The combination of resveratrol and LiCl showed effects on hWJ-MSC sheets similar to those of the basal chondrogenic medium. Adding Y27632 decreased both the proportion of hypertrophied cells and the expression of the hyaline cartilage markers. In vitro, DMSO was observed to impede the effects of the chondrogenic factors. The mouse knee defect model experiment revealed that hWJ-MSC sheets grown with the addition of resveratrol and Y27632 were well integrated with the surrounding tissues; however, after 3 months, the restored tissue was identical to that of the naturally healed cartilage injury. Thus, the combination of chondrogenic supplements may not always have additive effects on the progress of cell culture and could be neutralized by the microenvironment after transplantation.

## 1. Introduction

Wharton’s jelly mesenchymal stem/stromal cells (WJ-MSCs) have shown promising results in the field of regenerative medicine due to their differentiation potential and immunomodulatory properties [1]. WJ-MSCs can be isolated from the connective tissue surrounding the umbilical cord vessels, which makes them more primitive than any adult mesenchymal stem/stromal cells (MSCs); at the same time, their use does not cause the ethical problems that the use of fetal tissue does [2]. In addition to the adipogenic, chondrogenic, and osteogenic differentiation standards for MSCs, WJ-MSCs can also differentiate into endothelial cells [3], Schwann cells [4], endometrial cells [5], corneal epithelial cells [6], and other cell types; that is, they are able to transform into cells of other germ layers. Moreover, WJ-MSCs express pluripotency markers such as *Nanog*, *Oct-4*, and *Ssea-4*, although to a lesser extent compared with induced pluripotent stem cells (iPSCs) [7]. WJ-MSCs have immunomodulatory properties and express angiogenic and wound-healing factors [8]. Moreover, WJ-MSC-derived exosomes showed an anti-inflammatory potential, polarizing macrophages towards the M2 phenotype [9].

Due to their unique properties, WJ-MSCs are actively used for tissue regeneration, including that of hyaline cartilage, which is difficult to repair. As an example, the cocultivation of WJ-MSCs and costal chondrocytes in a pellet culture transplanted into rat joint defects led to an increase in the expression of cartilage markers and a decrease in hypertrophy [10]. The transplantation of human WJ-MSCs (hWJ-MSCs) within a chondroitin sulfate scaffold into a rat knee joint defect also resulted in defect filling, with the tissue containing high levels of proteoglycans and collagen type II [11]. A clinical study on the transplantation of WJ-MSCs seeded on a collagen matrix for the treatment of joint surface injuries demonstrated a filling of defects, although a qualitative analysis of the regenerated tissue was not performed [12].

Various agents that influence the signaling pathways of chondrogenesis were utilized to enhance chondrogenesis in MSCs in vitro. For example, it has been shown that adding lithium chloride (LiCl) to the chondrogenic medium stimulated the Wnt/β-catenin signaling pathway [13] and led to the increased synthesis of hyaline cartilage markers in monolayer and pellet cultures of hWJ-MSCs [14]. In this case, LiCl acts through the inhibition of GSK3 [15], which in turn inhibits the canonical Wnt signaling pathway due to β-catenin phosphorylation [16]. It has also been shown that Rho-associated protein kinase (ROCK) inhibition using a small molecule of Y2763 led to a more efficient retention of the chondrocyte phenotype in human chondrocyte pellet cultures [17] and increased chondrogenesis in human bone marrow-derived MSCs (BM-MSCs) [18], even in a monolayer culture of human chondrosarcoma cells [19]. Overall, ROCK inhibitors in 3D cultures can be used to improve the quality of aggregated cells and reduce cytoskeletal tension [20].

Resveratrol is a polyphenolic compound known for its antioxidant, anti-inflammatory, and anti-carcinogenic properties [21]. Among the targets of resveratrol, sirtuin 1 (SIRT1), adenosine monophosphate activated protein kinase (AMPK), and cyclooxygenase-2 are primarily noted [22]. For BM-MSCs [23], adipose MSCs [24,25], and WJ-MSCs, resveratrol improves the proliferation and further chondrogenic differentiation, as assessed by the expression of chondrogenic markers such as Sox9, collagen type II, and aggrecan.

In this study, we examined the benefits of different combinations of chondrogenic additives in culturing WJ-MSC sheets, since modifications in the cell culture media can enhance chondrogenesis. Resveratrol was added at the expansion stage, and the differentiation stage was supplemented with chondrogenic factors such as LiCl or Y2763. Next, we assessed the ability of the WJ-MSC sheets to restore defects in the articular cartilage in mice.

## 2. Materials and Methods

### 2.1. Human WJ-MSC Isolation

All the experimental protocols [26] were approved by the Local Ethical Committee of Sechenov University (Moscow, Russia) (no. 07-17 from 13 September 2017; no. 22-02 from 9 January 2023). Human WJ-MSCs from four donors were obtained from the biobank of Sechenov University (Moscow, Russia). Human umbilical cords were collected from the donors at full-term gestation with each mother’s consent for the research application. The human umbilical cord was placed into a 50 mL tube with DMEM/F12 (Gibco, Waltham, MA, USA) supplemented with 0.3 mg/mL of L-glutamine (Paneco-Ltd., Moscow, Russia) and 40 mg/mL of gentamicin (Paneco-Ltd., Russia) and taken to the laboratory on ice. The cords were rinsed twice with phosphate-buffered saline (PBS, Paneco-Ltd., Russia) containing 1% penicillin–streptomycin (Gibco), 0.6% fluconazole (Vertex, Saint-Petersburg, Russia) and 50 ug/mL of gentamicin (Paneco-Ltd., Russia,). The umbilical cord Wharton’s jelly was shredded into 2 cm fragments. The cord’s vessels were removed to avoid endothelial cell contamination. The Wharton’s jelly parts of the cord were cut into pieces of about 1 mm^3^. The umbilical cord Wharton’s jelly was shredded and then enzymatically digested using 0.2% collagenase NB4 (Nordmark, Uetersen, Germany), 0.005% hyaluronidase (Thermo Fisher Scientific, Waltham, MA, USA), and 10% accutase (PanBiotec, Aidenbach, Germany) for 1–2 h at 37 °C to facilitate the detachment and loosening of the Wharton’s jelly into the culture medium. Mesenchymal cells were isolated from Wharton’s jelly by passing the tissue through a 70 µm filter followed by centrifugation (at 100 g). The sedimented cells were cryopreserved using the standard cryopreservation protocol until needed in the following experiments.

In our study, we used pooled patient-derived WJ-MSCs to reduce interindividual confounder-associated variation [27,28,29,30]. 

### 2.2. Compliance with the Established Minimal Criteria for MSCs

Wharton’s jelly-derived cells were characterized using flow cytometry at passage 0 (P0). The cell surface antigens generally expressed by MSCs included CD90, CD73, CD105, CD44, CD29, and CD146, and the presence of CD45, CD34, CD19, CD11b, CD31, and HLA-DR (all markers–BD Biosciences, Franklin Lakes, NJ, USA) was undesirable. Cells without antibodies were used as blank samples. Cells (105 cells per sample) were incubated with antibodies (1:100 dilution) for about 30 min. The washed cells were analyzed using flow cytometry (Cell Sorter Sony SH800S, Sony Biotechnology, San Jose, CA, USA).

To confirm the compliance with another criterion, differentiation of the monolayer culture of Wharton’s jelly-derived cells toward osteogenic, adipogenic, and chondrogenic lineages was initiated. Cells grown using a commercial StemPro differentiation kit (Gibco, Thermo Fisher Scientific, USA) for 3 weeks were fixed and stained with Alizarin red (Merck, Thermo Fisher Scientific, Karlsruhe, Germany), Oil Red O (Merck, Thermo Fisher Scientific, Germany), and Alcian blue (Merck, Thermo Fisher Scientific, Germany) dyes, revealing the osteogenic, adipogenic, and chondrogenic differentiations, respectively. Cells grown in a low-glucose (1.5 g/L) Dulbecco’s Modified Eagle Medium (DMEM, Gibco™ 41966-029, Thermo Fisher Scientific, USA) supplemented with 10% fetal bovine serum (FBS, HyClone, Logan, UT, USA) were used as a control.

### 2.3. Preparation of Synthetic P(NIPAM-co-NtBA)-Based Matrices

Cells were cultured on unique, 100 µm-thick matrices that become water-soluble at temperatures below 30 °C. The matrices were made of a thermoresponsive copolymer, poly(N-isopropylacrylamide-*co*-N-tert-butylacrylamide) (P(NIPAM-*co*-NtBA) (M_n_ = 137,000 g mol^−1^, Ð = 4.75; NIPAM:NtBA weight ratio in the copolymer = 86:14), synthesized according to the procedure described in [31]. This copolymer dissolves at a lower temperature than commercially available poly(N-isopropylacrylamide) (25 °C vs. 32 °C) and is characterized by higher hydrophobicity that facilitates cell detachment, both in the monolayer and sheet. The thermoresponsive copolymer matrices are nontoxic and neutral to cell cultures [32], and they do not affect the cell immunophenotype [33].

Fibrous electrospun scaffolds were fabricated from a 15 wt% solution of P(NIPAM-*co*-NtBA) in absolute ethanol using an ESR100D NanoNC electrospinning machine [32]. All the processing parameters were preliminarily optimized to obtain defect-free fibrous materials. The distance between the needle tip and a conductive dynamic collector was 15 cm. The prepared solutions were loaded into a plastic syringe with a G20 needle of a 0.5 mm diameter. The applied voltage was 25.5 kV, and the flow rate was set at 1.5 mL/h.

### 2.4. Cell Culture Techniques and the Formation of Cell Sheets

The formation of a cell sheet from hWJ-MSCs involved two basic steps: expansion and differentiation (Figure 1). Cell expansion was performed in a low-glucose DMEM containing 10% FBS (HyClone, Logan, UT, USA) and FGF2 (10 ng/mL; PeproTech, NJ, USA) for 5 passages (base medium). A 1 μM resveratrol (Merck, Thermo Fisher Scientific, USA) solution in dimethyl sulfoxide (DMSO) was added to enhance cell expansion and the subsequent chondrogenic differentiation. In studies, various agents are often dissolved in DMSO. According to the manufacturer, resveratrol should be dissolved in DMSO before use due to its low water solubility (<0.05 mg/mL). The Y27632 reagent is also more soluble in DMSO (160 mg/mL) than water (90 mg/mL).

We hypothesized that the antioxidant action of resveratrol [21] and ROCK inhibition could prevent chondrocyte dedifferentiation in vitro [17]. The expansion in a medium with an equal volume of DMSO (Sigma, St. Louis, MO, USA) was used as a control. As another control, hWJ-MSCs were also expanded in a base medium (BM) without additives.

Following the expansion, cells were seeded onto the electrospun matrices and cultured in a chondrogenic medium based on a high-glucose (4.5 g/L) DMEM containing 1% insulin–transferrin–selenite (1.2.006; BioloT, Beijing, China), a 1% MEM Non-Essential Amino Acids solution (11140-035, Gibco, Thermo Fisher Scientific, USA), 10^−7^ M dexamethasone (Sigma, Thermo Fisher Scientific, USA), 50 μM ascorbate-2-phosphate, and TGFβ3 (10 ng/mL; PeproTech, USA). In different groups, the medium included the ROCK inhibitor Y27632 (10 μM; Sigma, Thermo Fisher Scientific, USA), which can also inhibit a noncanonical Wnt/RhoA pathway [34] or the canonical Wnt-pathway activator, LiCl (5 μM) [14]. Since Y27632 and LiCl were dissolved in DMSO, the control sample medium contained DMSO. Cells that expanded in the base medium were differentiated in a standard medium without additives. Cells were cultivated on the matrices for 3 or 5 weeks. In each study group, five cell sheets were cultured. One-half of each cell sheet was utilized for RT-PCR analysis, while the remainder was used for sectioning and histological analysis.

The chondrogenic differentiation step was carried out using the thermoresponsive copolymer matrices attached to the edge of previously UV-sterilized silicone rings (with a diameter of 9.6 mm) in 24-well plates. After adding the differentiation medium, the plates were placed in a CO_2_-incubator overnight. On the following day, the medium was removed from the wells, and 1.5 × 10^5^ cells in 200 μL of the differentiation medium per well were seeded on the matrices. One day later, the medium’s total volume was increased to 1 mL per well.

### 2.5. Animals

Two-month-old male nude mice (NU–A/A Tyrc/Tyrc Foxn1nu/Foxn1nu) purchased from the Nursery for Laboratory Animals (Pushchino, Russia) were used in the experiments. The animal experiments were approved by the Local Bioethics Committee (protocol no. 22-02 from 9 January 2023) of Sechenov University (Moscow, Russia). In the animal experiments, the guidelines and ethical policy of the Federation of European Laboratory Animal Science Associations (FELASA) were followed. The animals were kept in a room with a 12 h light cycle and free access to fresh water and rodent chow.

### 2.6. Acute Knee Injury Model

The mice were anesthetized through intraperitoneal injections of Zoletil^®^100 (tiletamine and zolazepam, 30 μg/g, Virbac, Carros Cedex, France) and Rometar^®^ (xylazine, 30 μg/g, Bioveta, Ivanovice na Hane, Czech Republic). The access to the cartilaginous surface of the femoral epiphysis was carried out with a layer-by-layer medial parapatellar incision and the following retraction of the patella to the lateral side. To reduce the postoperative load, the defect was formed in the intercondylar fossa using a surgical power drill (diameter 0.3 mm). The defect area was washed with sterile saline to remove the tissue debris. Using microsurgical tweezers, the cell sheets were placed into the cartilage injury area, filling the defect. Due to the cell sheet’s adhesive properties, it became possible to achieve its fixation without the use of additional fixative methods. After returning the patella to its original position, the articular cavity was sutured with an absorbable suture material (Appendix A). The sutured skin incision was treated with a Betadine^®^ antiseptic solution (povidone–iodine, 10%). The animals were monitored after the surgery and their condition was evaluated using the pain and distress evidences approved by the FELASA [35]. After the surgery, the animals preferred not to move unless it was necessary on the first day. Three days after the surgery, the rats did not exhibit any signs of pain. Within a week, the animals began to move freely around the cage, and their mobility and activity did not differ from that of the intact group.

To monitor every step of the surgical procedure, three control groups were selected: intact, sham-operated, and injured animals. There were 5 animals in each group. The joint capsule of the sham-operated animals was surgically opened, washed, and sutured without damaging the cartilage.

### 2.7. Sample Preparation

To separate the grown cell sheets and thermoresponsive P(NIPAM-co-NtBA)-based matrices, the samples were transferred to a 35-mm Petri dish with Hank’s solution (Paneco-Ltd., Russia) and left at 4 °C for about 20 min until the total dissolution of the matrices. With a scalpel, the cell sheets were split in half. One half of a cell sheet was fixed in 10% neutral buffered formalin at +4 °C for 4 h, dehydrated in 30% sucrose in PBS solution for 24 h and frozen at −80 °C in the embedding medium for cryotomy (Tissue-Tek^®^ O.C.T. Compound, Sakura Finetek, Torrance, CA, USA) for optimal section quality. The other half of the cell sheet was placed into an IntactRNA solution (BC031, Evrogen, Moscow, Russia) and stored at −20 °C for several days for the subsequent gene expression analysis.

The mouse knee tissue samples from the in vivo experiment were fixed in 10% neutral buffered formalin at +4 °C overnight. The next step was to decalcify the samples in a 10% EDTA solution (Sigma, USA) for two days, then dehydrate them with 30% sucrose in PBS and freeze them in the embedding medium for cryotomy.

### 2.8. Immunofluorescence and Histology Staining

For the immunofluorescence staining, cryostat (HM525 NX, Thermo Fisher Scientific, USA) cell sheet sections (18 μm) and cryostat tissue sections (30 μm) were incubated with antibodies to detect the presence of SOX9 (HPA001758, Sigma, USA), collagen type I (MGH C11, Imtek, Moscow, Russia), collagen type II (PA1-26206, Invitrogen, Waltham, MA, USA), MEF2C (HPA00553, Sigma, USA), and OSX (PA5-40411, Invitrogen, USA). For the antibodies targeting SOX9, MEF2C, and OSX, citrate buffer retrieval (citrate buffer (pH = 6.0), at 95 °C for 5 min) was performed, but for the antibodies targeting collagen type I and type II, pepsin antigen retrieval (0.5% pepsin solution in 10 mM HCl, at 37 °C for 25 min) was more effective.

After the immunofluorescence staining analysis was finished, the cover glasses were carefully removed, and the sections were stained with toluidine blue/fast green to assess the abundance of proteoglycans in the matrix.

### 2.9. Gene Expression Analysis

By using the ExtractRNA reagent (BC032, Evrogen, Russia), we extracted total RNA from cell sheets using the phenol–chloroform method. After that, the RNA samples were treated with the DNAse E reactant (EK007S, Evrogen, Russia) to improve their quality. The RNA purity and concentration were quantified using a NanoPhotometer^®^ N60 microvolume UV spectrophotometer (Implen, München, Germany). The obtained RNA was converted to the complement DNA (cDNA) using the MMLV reverse transcription kit (SK021, Evrogen, Russia) in accordance with the manufacturer’s recommendations, and all samples were diluted to a 200 ng/μL concentration. In each reaction, 600 ng of cDNA was added to the qPCRmix-HS SYBR buffer (PK147, Evrogen, Russia) in 25 μL of the final volume. Using the Primer-BLAST (NCBI, USA) freeware, the primer pairs were designed (Appendix A) according to standard PCR guidelines. A DTprime real-time PCR instrument (DNA-Technology, Russia) was set up to perform qRT-PCR, including the denaturation (95 °C), annealing, and extension (72 °C) steps. The annealing temperature was individual for each primer pair (between 57 °C and 63 °C). The reactions were generally run for 45 cycles with the addition of a melt curve (dissociation curve) to exclude the presence of nonspecific products. Each of the PCR reactions included three technical replicates. Endogenous reference genes, such as those encoding glyceraldehyde phosphate dehydrogenase (GAPDH, NCBI Gene ID 2597) and ribosomal protein L13a (RPL13A, NCBI Gene ID 23521), were used to normalize the transcriptional profiles of the samples.

### 2.10. Statistical Analysis

For in vitro experiments, the data are presented as the mean and 95% confidence intervals. The Shapiro–Wilk test confirmed the normal distribution of the data. One-way ANOVA with Tukey’s multiple-comparison test was used for the comparison of multiple groups. *p*-values < 0.05 were considered significant.

The real-time PCR data are quantified relatively. The Ct of the target genes was normalized to the Ct of the reference genes encoding 60S ribosomal protein L13a (RPL13A, NCBI Gene ID 23521) and glyceraldehyde-3-phosphate dehydrogenase (GAPDH, NCBI Gene ID 2597). Each group included 3–5 cell sheet samples. The ΔΔCt model was applied to calculate the ratio of the target gene expressions [33]. For the group comparisons in the statistical analysis, a factorial ANOVA was used [34] (the Ct was used as the dependent variable; the grouping variables “group”, “cell sheet number”, and “gene” were assigned as fixed factors in the model). The statistical analysis was performed on the IBM^®^ SPSS^®^ 17.0 software platform. *p*-values < 0.01 were considered significant. The changes in the target gene relative expression ratio of less than 30% compared to the control group were declared nonsignificant (having no physiological significance) regardless of the statistical test result.

## 3. Results

### 3.1. WJ-MSCs’ Phenotypic Characterization

According to single-cell RNA sequencing, BM-MSCs and WJ-MSCs were found to be heterogeneous mixtures of cell subpopulations [36]. The isolation technique appeared to be responsible for the differences in the characteristics of MSCs isolated by trypsinization, and it was not possible to achieve a 99% population purity using the method mentioned in [37]. Thus, we believe that the characterization of the resulting populations is an essential aspect of any MSC research. In the previous studies, the minimal MSC criteria for WJ-MSCs were assessed at later passages [38,39]. In the present study, we demonstrate that MSCs can be obtained as early as the first passage [40].

MSCs were derived from Wharton’s jelly of the human umbilical cord from four donors, pooled, and characterized using flow cytometry for the MSC marker profile [40]. The isolated cell population’s characterization (Figure 2A) excluded the presence of D34, CD19, and CD11b markers of blood and hematopoietic cells (Figure 2B–D), CD31-positive endothelial cells (Figure 2E), and HLA-DR immune cells (Figure 2G). More than 99% of the population was positive for the MSC markers CD90, CD29, CD44, CD73, and CD105 (Figure 2G–K), but only half of the cells expressed CD146 (Figure 2L). In our experiments, besides the CD146-positive subpopulation, there were also CD146-negative WJ-MSCs. The presence of such MSCs is generally accepted since CD146-negative cells have been described in the bone marrow, representing a distinct subpopulation of MSCs [41].

To confirm the compliance with the MSC criteria, it was demonstrated that hWJ-MSCs could be differentiated into three mesenchymal lineages in the specialized commercial media (Figure 2M–O).

### 3.2. Assessment of the Differentiation Grade of Cell Sheets after 3 and 5 Weeks of Culturing in the Chondrogenic Medium

The general scheme for the group distribution based on the hWJ-MSC sheet culturing methods is shown in Figure 1. To improve the chondrogenic differentiation, hWJ-MSCs are recommended to be expanded in a medium supplemented with resveratrol, according to the published data [23,42]. Since resveratrol was dissolved in DMSO, the group expanded in the BM group supplemented with DMSO (DD group) was used as one of the control groups. The cell proliferation rate was not affected by resveratrol or DMSO compared to the BM group, and the cumulative numbers of cells were similar at each passage (Figure 2P). After the fifth passage, the cells were grown on synthetic P(NIPAM-co-NtBA)-based matrices in the chondrogenic medium for 3 weeks, forming 3D sheet-like structures. Two groups selected for the in vivo experiment were grown for 5 weeks. We gathered the most successful strategies from the literature to enhance the chondrogenic differentiation of hWJ-MSCs to compare their efficacy. We supplemented the chondrogenic medium with Y27632 [20], an inhibitor of ROCK and a noncanonical Wnt-signaling pathway, or with LiCl, the canonical Wnt-signaling pathway activator [14].

The presence of chondrogenic markers such as SOX9 transcription factor (Figure 3a–e) and collagen type II (Figure 3a′–e′), as well as the bone and fibrous tissue marker collagen type I (Figure 3a″–e″) was observed in the cell sheet sections using immunofluorescence staining and RT-PCR. The toluidine blue staining failed to reveal proteoglycans in the extracellular matrix of the cell sheets (Figure 3A–E). MEF2C, the hypertrophy marker, was found in almost all the cell sheets, except for the resveratrol+Y27632 (RY) and resveratrol+LiCl (RL) groups (Figure 3a‴–e‴). OSX, the osteocyte marker, was not identified after 3 weeks of culture (Figure 3a⁗–e⁗).

Neither group showed significant differences in the collagen type I expression compared to the BM group after 3 weeks in vitro. The resveratrol combined with LiCl (RL group) demonstrated an influence comparable to that of the BM group, with all four markers used to assess the degree of cell sheets’ chondrogenic differentiation (Figure 3F). In 3 weeks, the RY and RD groups did not show an enhanced chondrogenic effect compared to the BM group, but their hypertrophic responses (MEF2C and COL1A1 expression) were decreased (Figure 3F).

It is worth noting that the DD group had a significant difference in the chondrogenic characteristics compared to the DMSO-free BM group after 3 weeks of culturing. According to the RT-PCR data, DMSO had a deterrent effect on the chondrogenic differentiation of hWJ-MSC sheets (Figure 3F,G). Thus, we can conclude that the final effect of the culture media on the differentiation process represented a cooperative effect of the supplements and their solvent, DMSO.

Since no significant effect on the chondrogenic sheet differentiation was observed in the RL group compared with the BM group, and the effect in the RD group was similar to that in the RY group, the differentiation stage of the RY group and the control groups (DD and BM) was extended to 5 weeks. We took into account a recent publication demonstrating that prolonging the differentiation period can enhance the chondrogenicity of cultures [14]. After prolonging the differentiation stage in vitro to 5 weeks, the immunofluorescence staining of cell sheet sections failed to reveal significant changes: all cell sheets exhibited SOX9-positive cells (Figure 4a–c) and were well-stained for collagen types II and I (Figure 4a′–c′,a″–c″), including some MEF2C-positive cells (Figure 4a‴–c‴). However, the results of staining for OSX and proteoglycans were negative after 5 weeks of differentiation (Figure 4a⁗–c⁗,A–C). The RY group did not show significant changes in the expression of SOX9, COL2A1, and MEF2C genes compared with the BM group (Figure 4D). However, the DD group showed a trend toward a decreased expression of the matrix proteins collagen types I and II (Figure 4D). Comparison of the RY group with the DD group (Figure 4E), allowing to exclude the DMSO impact, showed chondrogenic positive changes in the RY group, including a significant increase in the expression of the collagen type II gene and a significant decrease in the expression of the gene encoding MEF2C. We observed a unidirectional effect on hWJ-MSC cultures in the BM and the RY groups compared to the DD group (Figure 4E) after prolonging the differentiation stage to 5 weeks.

### 3.3. Estimation of the Regenerative Abilities of hWJ-MSC Sheets in a Rat Model of a Cartilage Defect

To estimate the regenerative ability of the prepared hWJ-MSC sheets in a model of acute defect, the RY and DD groups were selected, since after 5 weeks they showed the least tendency towards fibrocartilage. The in vivo experiments were conducted on nude mice due to their low immune response to xenograft transplantation [43]. The cell sheets were grown in the chondrogenic medium for 5 weeks, since these were denser and more convenient for surgery manipulations. The analysis of the cartilage defect regeneration was performed at 3 weeks and 3 months after the cell sheet transplantations through the histological and immunofluorescence staining of tissue sections (Figure 5).

Three weeks postoperation, single SOX9-positive cells were still found in the transplanted cell sheets (Figure 5D,E) for both the DD and RY groups, and staining for collagen type I was observed (Figure 5D″,E″). The cell sheets were well integrated into the underlying tissue (Figure 5D‴,E‴), but did not restore the cartilage structure and generally did not differ significantly from the control group with defects (Figure 5C–C‴).

After 3 months, SOX9 staining was no longer detected in the transplanted sheets (Figure 5I,J), while the staining for collagen type I was much more explicit compared to the 3-week time point (Figure 5I″,J″). Collagen type II (Figure 5I′,J′) and proteoglycans (Figure 5I‴,J‴) were not detected through staining. The transplanted cell sheets filled the cartilage defects and integrated into the surrounding tissues, but the histological staining results did not demonstrate any advantages compared to the control group with defects (Figure 5H–H‴).

Thus, despite the results obtained in vitro, the transplanted hWJ-MSC sheets did not significantly improve the healing process of the joint cartilage defects at distant time points. Overall, one should not ignore the fact that the use of immunodeficient animals allows for avoiding inflammation and immune rejection following the xenotransplantation, but on the other hand, it leads to the negative effects by excluding some types of cells from the repair process [44].

## 4. Discussion

Our investigation focused on the chondrogenic properties of 3D cell sheets from a hWJ-MSC culture with the use of specific agents to enhance the chondrogenic differentiation: resveratrol at the expansion stage [23], and the canonical Wnt pathway activator LiCl [14] or the ROCK inhibitor Y27632 [20] at the differentiation stage.

As shown by the studies, each of the mentioned reagents has certain advantages in the chondrogenic differentiation: the stimulation of SOX9 synthesis, increase in the production of hyaline cartilage matrix proteins, and reduction of hypertrophy. However, we did not observe a synergistic effect when resveratrol was applied during the expansion stage in combination with the reagents promoting the chondrogenic differentiation of WJ-MSCs. The application of resveratrol during the expansion stage and LiCl during the differentiation stage did not improve the chondrogenesis of the grown WJ-MSC sheets compared with the BM group. We observed a statistically significant decrease in the expression of a chondrogenic marker, Sox9, in the RL and RY groups.

The sequential processes of inhibition (by resveratrol) and activation (by LiCl) of the canonical Wnt pathway or ROCK suppression (by Y27632) likely induced a complex response of WJ-MSCs to the activation of chondrogenic differentiation initiated by TGFβ3. Resveratrol, Y27632, and LiCl are agents that have been reported to have chondrogenic effects, but their actions are complex and have not been well researched. Resveratrol improved MSC proliferation and further chondrogenic differentiation by stimulating the ROCK inhibitor SIRT1 [25] and inhibiting the Wnt/β-catenin pathway [45] (without Wnt activators during differentiation [42]). Nevertheless, the effects of the ROCK inhibition on the MSC chondrogenesis differed substantially depending on the culture models [18]. The use of ROCK inhibitors generally increased the expression of the chondrocyte markers in a monolayer culture [19,46], but led to hypertrophy [18]. At the same time, the cultivation of differentiated MSCs with LiCl decreased the hypertrophy, according to the published data [14]. LiCl stopped the GSK3 protein from the phosphorylation of β-catenin and activated the canonical Wnt/β-catenin signaling pathway. Thus, LiCl should act synergistically with TGF-β3 in the chondrogenic differentiation of monolayer and pellet hWJ-MSC cultures [14]. It is worth noting that the Wnt/β-catenin signaling pathway often crosstalks with other signaling pathways, which interact with each other to modulate chondrogenesis and cartilage development [47]. As well, the ROCK inhibition using a small molecule of Y2763 led to a more efficient retention of the chondrocyte phenotype in human chondrocyte pellet cultures [17] and increased chondrogenesis in human MSCs in the pellet culture, upregulating chondrogenic signaling [18]. 

The interactions between chondrogenic agents during MSC culturing are complex, multicomponent, and frequently overlap. They have not been sufficiently studied to allow for a clear understanding of the resulting effects. Apparently, the long-term pretreatment of cell cultures with resveratrol could affect their state and the subsequent activation of the Wnt pathway at the differentiation stage. This assumption is generally consistent with the fact that hyperactivation of Wnt can cause the suppression of chondrogenesis [48]. Overall, there is no consensus about the canonical Wnt effect on the chondrogenic differentiation of cells, since it can have different effects on different cell types [49].

As well, the WJ-MSC population’s heterogeneity may explain the difference between the results of differentiation by the chondrogenic agents. Both the tissue source and extraction method have a significant impact on the final result of chondrogenic differentiation [50]. The MSC populations extracted also contain diverse subpopulations, including cells that have already started the differentiation process [51]. Overall, MSC subpopulations have shown great heterogeneity at the transcriptome level [36], and they are capable of preferring certain cell lineages to differentiate. The method of cell extraction also contributes to the heterogeneous nature of the MSC population, especially since achieving a 99% purity of the isolated population using the enzymatic isolation method is challenging [52]. In addition, it is essential to remember that MSC donors are diverse in terms of their age, gender, and health [50].

Surprisingly, we also found a nonzero effect of DMSO on the chondrogenic differentiation of WJ-MSCs sheets. Notably, the changes in the marker expression profiles had a very similar direction in restraining the chondrogenic differentiation in the RY, RD, and DD groups compared to the BM group. One may assume that the influence of DMSO on the WJ-MSC cultures exceeded the effect of LiCl and Y27632 combined with resveratrol. Another possibility is that the chondrogenic reagents in the study negated the effects of each other when applied in combination, thereby identifying the DMSO influence on the gene expressions in WJ-MSCs during the chondrogenic differentiation. Apparently, the interaction of chondrogenic supplements was complex in WJ-MSCs.

The effect of DMSO, applied as a solvent for the tested agents, on cell differentiation has been demonstrated in the previous publications. For example, it was observed on umbilical cord MSCs when studying the effects of Notch activation [53]. Overall, the effects of DMSO can include the stimulation of lipid metabolism [54] and metabolism in general [55], enhancement of hepatocyte differentiation [56], and even changes in epigenetic profiles [57]. It is unclear how exactly DMSO affects cell differentiation in our experiments, but its effect is quite noticeable.

The transplanted cell sheets of the RY and DD groups were clearly visualized both at 3 weeks and 3 months. Good integration with the surrounding tissues was indicated, but the transformation of cell sheets into the native hyaline cartilage structure did not occur. While we found single SOX9-positive cells in the sheets after 3 weeks, we did not observe signs of chondrogenic differentiation after 3 months. As was the case with other approaches [58], we also observed a dedifferentiation process in vivo due to the instability of the chondrogenic cell phenotype [59]. The tissue microenvironment that developed in the damaged and inflamed joint negatively affected cell differentiation [60] in a sheet and led to explicit fibrosis of the damaged cartilage region.

## 5. Conclusions

WJ-MSCs can be a promising source of cells for regenerative medicine due to their unique properties that distinguish them from other MSCs. Their application in clinical practice is complicated by their heterogeneity and phenotypic instability, similar to other types of MSCs. In this study, we did not obtain convincing evidence for the benefits of using combinations of supplements such as resveratrol, Y27632 (a ROCK inhibitor), and LiCl when attempting to direct WJ-MSCs to chondrogenic differentiation in 3D sheets grown on synthetic P(NIPAM-co-NtBA) matrices. The concentrations of the chondrogenic agents that were previously studied were applied to WJ-MSCs in the present work, and the standard cultivation period was chosen. Despite our efforts, there were no significant chondrogenic effects revealed. The heterogeneity of WJ-MSCs and the 3D culture model selected played a major role in the results obtained. It is also worth noting that analyzing chondrogenic markers can provide admissible effects; however, the interpretation of the chondrogenic differentiation results could be distorted when the data are evaluated without the analysis of the hypertrophy markers.

Based on the present study, chondrogenesis stimulation with combinations of supplements such as resveratrol, Y27632, and LiCl cannot be appreciated as a potential technique to reveal the regenerative potential of WJ-MSCs in the treatment of joint cartilage injuries. In addition, due to the evident negative effect of DMSO, we recommend avoiding the use of DMSO during chondrogenic differentiation of MSCs to eliminate the contingent impact of the reagent.

## Figures and Tables

**Figure 1 cells-12-02845-f001:**
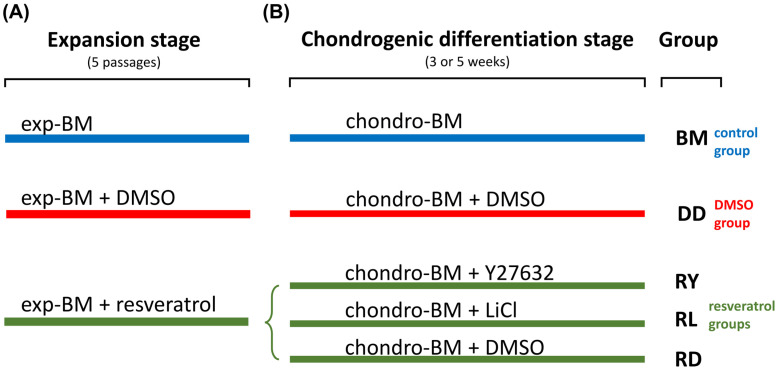
The design of the in vitro experiment. After 5 passages, the expansion stage was completed (**A**) and followed by 3 and 5 weeks of chondrogenic differentiation (**B**); exp-BM—expansion base medium; chondro-BM—chondrogenic differentiation base medium; DMSO—dimethyl sulfoxide; LiCl—lithium chloride. Each study group included five cultured cell sheets.

**Figure 2 cells-12-02845-f002:**
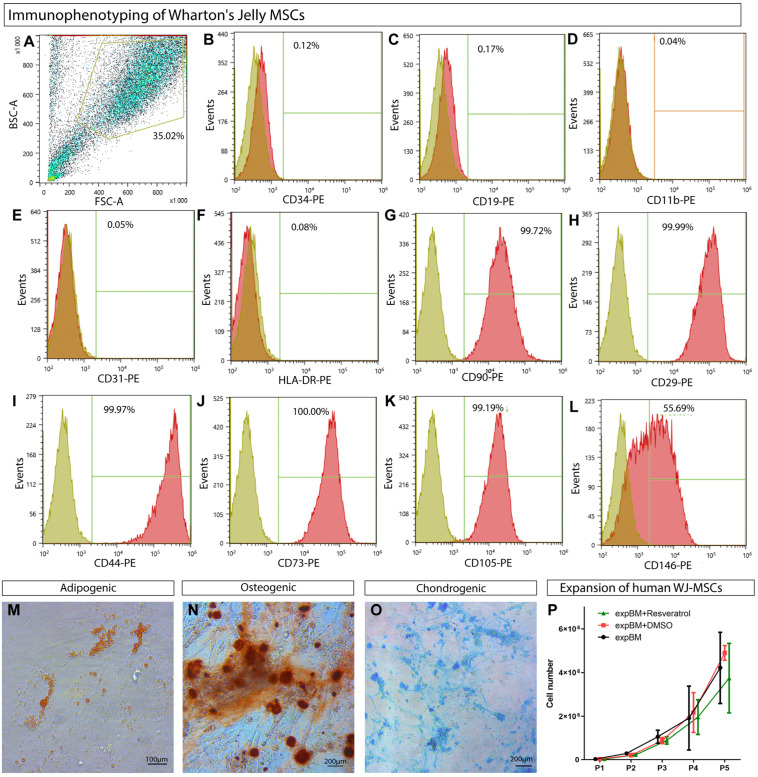
Immunophenotype analysis of hWJ-MSCs. Characterization of cell phenotype using flow cytometry (**A**). The human Wharton’s jelly-derived cells were stained with antibodies against the negative MSC markers CD34, CD19, CD11b, CD31, and HLA-DR (**B**–**F**), and the positive MSC markers CD90, CD29, CD44, CD73, CD105, and CD146 (**G**–**L**). Sand color indicates the negative control, red indicates the marker. Three-lineage differentiation of hWJ-MSCs to adipogenic (**M**), osteogenic (**N**), and chondrogenic (**O**) lineages. The diagram of cumulative number of cells proliferated (**P**); *n* = 6 (data obtained from 2 independent experiments, 3 replicates each); exp-BM—expansion base medium; chondro-BM—chondrogenic differentiation base medium.

**Figure 3 cells-12-02845-f003:**
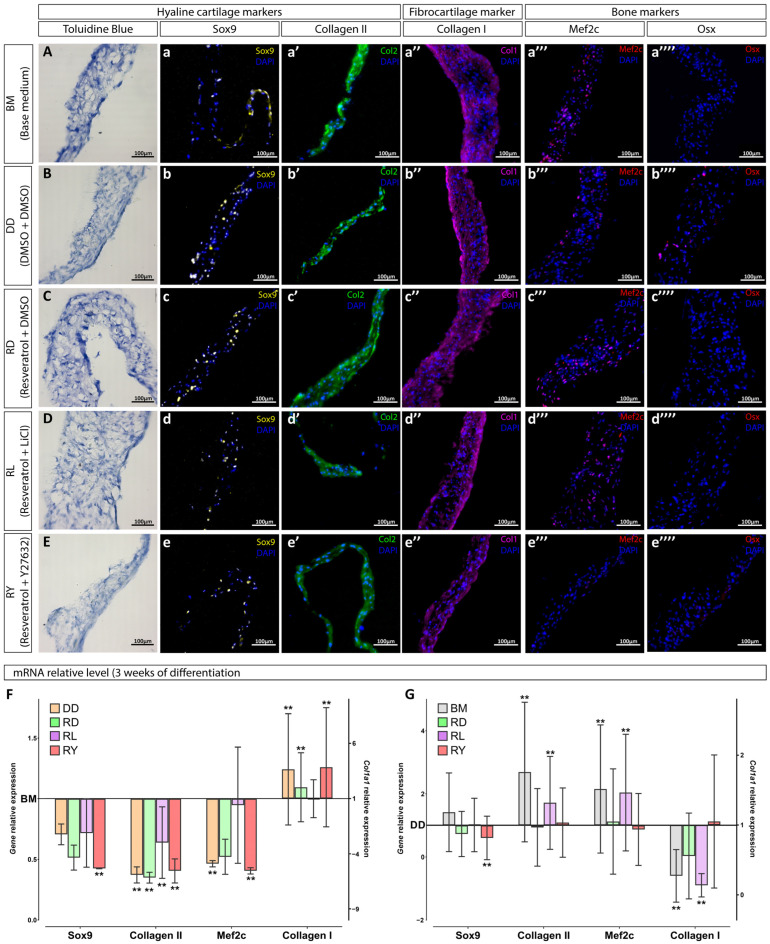
Characterization of hWJ-MSC sheets after 3 weeks in the chondrogenic medium. The histological staining for proteoglycans was performed with toluidine blue (**A**–**E**). Immunostaining with antibodies against the chondrocyte markers, SOX9 (**a**–**e**) and collagen type II (**a′**–**e′**); a fibrocartilage marker, collagen type I (**a″**–**e″**); and hypertrophy and osteocytes markers, MEF2C (**a‴**–**e‴**) and OSX (**a⁗**–**e⁗**), respectively. Relative expression of mRNA in cell sheets after 3 weeks of differentiation (**F**,**G**) displayed relative to mRNA levels in BM (base medium) group (**F**) and DD group (**G**); *n* = 5 (data obtained from 2 independent experiments, 3 and 2 replicates each); **—*p*-value < 0.01.

**Figure 4 cells-12-02845-f004:**
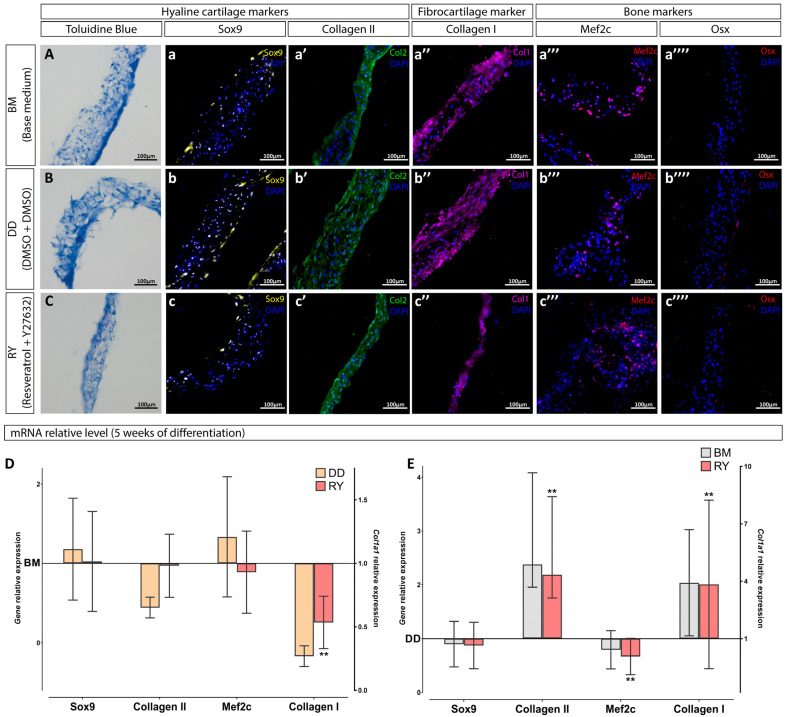
Characterization of hWJ-MSC sheets after 5 weeks in the chondrogenic medium. The histological staining for proteoglycans was performed using toluidine blue (**A**–**C**). Immunostaining with antibodies against the chondrocyte markers, SOX9 (**a**–**c**) and collagen type II (**a′**–**c′**); a fibrocartilage marker, collagen type I (**a″**–**c″**); and hypertrophy and osteocytes markers, MEF2C (**a‴**–**c‴**) and OSX (**a⁗**–**c⁗**), respectively. Relative expression of mRNA in cell sheets after 5 weeks of differentiation (**D**,**E**) displayed relative to mRNA levels in BM (base medium) group (**D**) and DD group (**E**). *n* = 5 (data obtained from 2 independent experiments, 3 and 2 replicates each). **—*p*-value < 0.01.

**Figure 5 cells-12-02845-f005:**
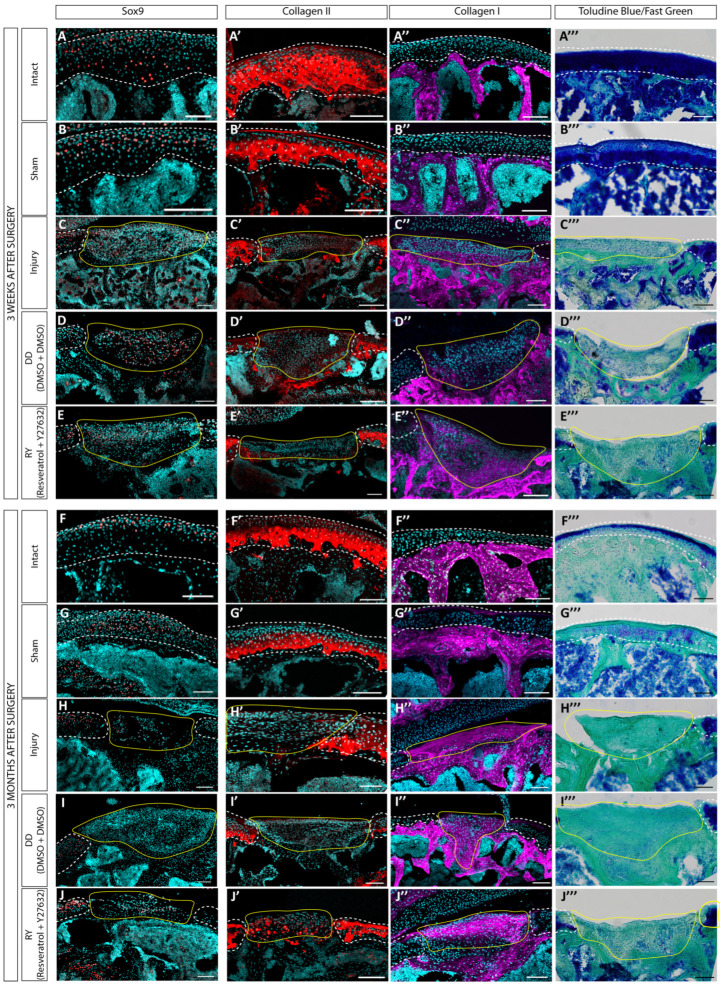
Immunostaining and histology analysis of joint sections after the transplantation of hWJ-MSC sheets. Assessment of the transplanted hWJ-MSC sheets at 3 weeks (**A–E,A′–E′,A″–E″,A‴–E‴**) and 3 months after the surgery (**F**–**J,F′–J′,F″–J″,F‴–J‴**). Immunostaining with antibodies against the chondrocyte markers, SOX9 (**A**–**J**) and collagen type II (**A′**–**J′**) and a fibrocartilage and bone marker, collagen type I (**A″**–**J″**). The histological staining was performed with toluidine blue/fast green (**A‴**–**J‴**). Representative images of study groups with 5 samples are shown. The damage area is highlighted with a yellow line, intact cartilage is highlighted with a white dotted line. All scale bars correspond to 100 μm.

## Data Availability

The data that support the findings of this study are available from the corresponding author upon reasonable request.

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
