# Peer review of "Resveratrol’s Impact on the Chondrogenic Reagents’ Effects in Cell Sheet Cultures of Wharton’s Jelly-Derived MSCs"

_cells, 2023, doi:10.3390/cells12242845_

Round 1

Reviewer 1 Report

Comments and Suggestions for Authors

The authors of the article conducted a significant experiment of interest in the development of tissue-engineered constructs for the repair of joint cartilage injuries. Studying the benefits of different combinations of chondrogenic additives in culturing WJ-MSC sheets have been reported.

However, it is not clear what these benefits are. It is necessary to indicate this more clearly in the abstract and conclusions.

In addition, there are technical inaccuracies (in some places "FBS" is indicated instead of "PBS"; in reference 33 "Wang, T. yi" is indicated instead of "Wang, T.-Y.").

Author Response

We would like to express our gratitude for the positive assessment of our work and for taking the time to read our research.

Previous studies have indicated that each of the chondrogenic additives, such as resveratrol, LiCl, and Y27632, improves and facilitates the differentiation process of WJ-MSCs (Choi et al., 2018; Tanthaisong et al., 2017; Wang et al, 2018). We used these additives in different stages of cell culturing in this study, but we didn't see any synergistic effects in the results. The effect of chondrogenic stimulation on WJ-MSC cultures was slightly different between basic chondrogenic differentiation medium, which contains growth factor Tgfb3, and chondrogenic medium supplemented with agents. Moreover, the slight difference was completely equalized by the in vivo experiment. The abstract has been revised and potential explanations for the observed results have been included in the discussion. We hope that the present manuscript has become more transparent and understandable.

We appreciate your important note about technical inaccuracies. Therefore, we have clarified the abbreviation of the cultural supplement FBS (fetal bovine serum) in the methods section (page 3, line 122).

We have corrected the author's name misspelling in the bibliography (page 14, line 644).

Choi, S.M.; Lee, K.-M.; Ryu, S.B.; Park, Y.J.; Hwang, Y.G.; Baek, D.; Choi, Y.; Park, K.H.; Park, K.D.; Lee, J.W. Enhanced Articular Cartilage Regeneration with SIRT1-Activated MSCs Using Gelatin-Based Hydrogel. Cell Death Dis. 2018, 9, 866,

Tanthaisong, P.; Imsoonthornruksa, S.; Ngernsoungnern, A.; Ngernsoungnern, P.; Ketudat-Cairns, M.; Parnpai, R. Enhanced Chondrogenic Differentiation of Human Umbilical Cord Wharton’s Jelly Derived Mesenchymal Stem Cells by GSK-3 Inhibitors. PLoS One 2017, 12

Wang, K.-C.; Egelhoff, T.T.; Caplan, A.I.; Welter, J.F.; Baskaran, H. ROCK Inhibition Promotes the Development of Chondrogenic Tissue by Improved Mass Transport. Tissue Eng. Part A 2018, 24, 1218–1227

Reviewer 2 Report

Comments and Suggestions for Authors

I have read and reviewed this article. An interesting study has been conducted. However, the results are not expected. I suggest to rewrite the text in terms of English writing. In some cases, such as lines 117, 131, and 133, it is necessary to correct the numbers and writings. The reason for the difference in the number of study groups in different experiments should be explained. For example, why has the RL group not been investigated invivo?

Comments on the Quality of English Language

 I suggest to rewrite the text in terms of English writing. In some cases, such as lines 117, 131, and 133, it is necessary to correct the numbers and writings. 

Author Response

Your thorough review of our research and the significant comments you made are greatly appreciated.

Prior to the start of the in vivo experiment Groups of cell sheets were analyzed 3 and 5 weeks after starting cultivation (each group included 5 sheets). At each stage, we discarded groups of cell sheets showing the least degree of chondrogenic transition, and we excluded one of the groups that demonstrated similar effects. By doing this, we were able to reduce the number of animal groups involved in the in vivo experiment (5 animals in each group).

Groups RL and RD were excluded at different stages of the experiment. At 5 weeks the RL group showed no differences compared to the control BM group. The RD (including resveratrol and DMSO) and RY (including resveratrol, DMSO, and Y27632) groups had similar effects on cell sheet differentiation, which means that the Y27632 molecule had no impact on the differentiation process (page 7, lines 340-342).

The analysis of 5 weeks' cell sheets led to the selection of the RY and DD groups for the in vivo experiment. The expression of the fibrotic marker collagen type I was decreased in the RY group compared to the control BM group (Fig4, D), while the expression of this marker was even lower in the DD group (Fig4, E). However, the expression of the chondrogenic marker collagen type II was higher in the RY group compared to DD (Fig4, E). When it comes to the injury model, transplanting cell sheets from the RY and DD groups could have more beneficial effect compared to the other cell sheets groups.

Relevant explanations have been added to the text on page 9, lines 394-395.

The language in the text has been corrected (lines 51, 52, 56, 59, 69, 70, 77, 78, 80, 115, 131, 133, etc.).

Reviewer 3 Report

Comments and Suggestions for Authors

In the manuscript: “Resveratrol Impact on the Chondrogenic Reagents’ Effects in Cell Sheet Cultures of Wharton's Jelly-Derived MSCs”, the authors discussed about the role of resveratrol during the cell expansion stage and as addition to Wnt pathway activator, LiCl, and a Rho-associated protein kinase inhibitor, Y27632, during the differentiation stage on  chondrogenic differentiation of hWJ-MSCs in 3D sheets.

Overall, this manuscript results very interesting, the authors clearly explain the rational of the study and discussed the topic point by point.

However, we would like to invite the authors  to clarify some critical points:

1.       Please check the check punctuation and spaces;

2.       The abstract is not clear, in particular the experiments performed;

3.       Page 2, lines 67-71 “Various agents that influence the signaling pathways of chondrogenesis were utilized to enhance the chondrogenesis in MSCs in vitro. For example, lithium chloride was shown to inhibit GSK-3 [13], to inactivate phosphorylation of β-catenin and initiate the Wnt signaling pathway [14]. It had been shown that adding LiCl to the chondrogenic medium stimulated the Wnt/β-catenin signaling pathway [15] and led to the increased synthesis of hyaline cartilage markers in monolayer and pellet cultures of hWJ-MSCs [16]”. Please, better describe the cited pathway;

4.       Page 2, lines 79-80 “Resveratrol, a polyphenolic compound, increased the expression of extracellular matrix proteins in WJ-MSCs when added to the chondrogenic medium [21]”. Please, better introduce this compound and its beneficial properties;

5.        Please, try to better organize the discussion, focus on your results with respect to the available data in literature.

Comments on the Quality of English Language

minor mistakes of spelling are here present

Author Response

We appreciate your high evaluation of our work and your attentive approach to the study material. We are prepared to address your comments in detail.

  1. We have examined the punctuation and spaces and made corrections in the lines 51, 52, 56, 59, 69, 70, 77, 78, 80, 115, 131, 133, etc.
  2. We attempted to comply with your suggestion and meet the journal's requirement for a 200-word abstract at one time. We are hopeful that the changes have made Abstract more meaningful (lines 15-29).
  3. We appreciate you pointing out the confusion in the reasoning. According to the literature, β-catenin is inactivated through phosphorylation by the GSK3 protein, which results in the inactivation of the canonical Wnt/β-catenin pathway. Inhibition of GSK3 by LiCl activates the Wnt/β-catenin signaling pathway. The text now includes clarifications (page 2, lines 62-64)
  1. Thank you very much, we really haven't provided much information about resveratrol. In the Introduction, we have included text that provides information about this reagent (page 2, lines 70-75).
  2. The Discussion section has been updated with more literature data on the discussed topic (page 9, lines 437-440; page 10, lines 444-461, lines 470-480).

Reviewer 4 Report

Comments and Suggestions for Authors

The authors used resveratrol during the cell expansion stage. During the differentiation stage, they added lithium chloride (LiCl) and a Rho-associated protein kinase inhibitor, Y27632, to investigate their effects on the chondrogenic differentiation of hWJ-MSCs in 3D sheets. They also carried out a pre-clinical trial in mice to test in vivo the impact of possible differentiation after using resveratrol, DMSO, LiCl, and Y27632. The use of resveratrol did not enhance the chondrogenic differentiation. Despite all efforts and well-conducted research, null or similar results using resveratrol associated with LiCl and Y27632 were found compared with the basal chondrogenic medium for differentiation. In the pre-clinical study, the restored tissue did not differ from the cartilage injury, which healed naturally after three months.

Therefore, the manuscript would help to guide some paths that should be avoided in future studies involving MSC associated with resveratrol, DMSO, LiCl, and Y27632.

INTRODUCTION

Line 68: Enter the abbreviation for lithium chloride (LiCl)

MATERIALS AND METHODS

1-    Standardize so that all reagents are standardized with brand and origin. Several of them throughout the text do not have this information.

2-    In the acute knee injury model, it was unclear how many animals were used in total nor how many were used in the group that finished in three weeks and in the group that finished in three months. It was also not detailed how the tissue was removed for immunofluorescence analysis and histology staining, as well as the euthanasia of the animals.

3-    As shown in the diagram in Figure 1Q, the groups should be presented in the methodology to facilitate understanding of the experiments. A schematic drawing of the study or a graphic abstract at the beginning of the method would be very enlightening.

4-    Was only one concentration of resveratrol used? Testing different concentrations of resveratrol could be more informative. Including a control group trying the other supplements in a micromass culture without using ad matrices could also exclude interference from matrix compounds.

5-    Carrying out clinical assessments regarding the animals' pain, behavior, and walking before and after surgery could provide additional data for the evaluations.

RESULTS

1-    In Figure 1P, chondrogenic differentiation is not visible. The photo will be presented at a higher magnification to highlight cuboidal cells and gaps around the young chondrocytes.

2-    Figure 1Q should be presented in more detail in the methodology.

3-    Many of the comments in this section could be in the discussion.

DISCUSSION

In the introduction, the authors mention two successful ones in increasing proliferation and differentiation when they added resveratrol to the chondrogenic differentiation medium. Likewise, they cite successful experiments in chondrogenic differentiation using LiCl and Y2763. Should the authors better explore the studies mentioned and others I suggest below to create hypotheses about why the research cited succeeded in increasing chondrogenesis and why the research presented in this manuscript did not? Could it be due to the concentration of the reactants? Or by the solvent used to dissolve the DMSO? Or by the different sources of MSC? Or by the evaluation time? Or could the combination of so many variables have interfered with the final result? Or were the evaluations of other works, not the most appropriate or not as accurate? Could a control without using the matrix, such as micromass differentiation, have helped? Finally, if better discussed, there are many points that could add more value to the manuscript.

Keshavarz G, Jalili C, Pazhouhi M, Khazaei M. Resveratrol Effect on Adipose-Derived Stem Cells Differentiation to Chondrocyte in Three-Dimensional Culture. Adv Pharm Bull. 2020 Jan;10(1):88-96. doi: 10.15171/apb.2020.011. Epub 2019 Dec 11. PMID: 32002366; PMCID: PMC6983992.

Peltz L, Gomez J, Marquez M, Alencastro F, Atashpanjeh N, Quang T, Bach T, Zhao Y. Resveratrol exerts dosage and duration dependent effect on human mesenchymal stem cell development. PLoS One. 2012;7(5):e37162. doi: 10.1371/journal.pone.0037162. Epub 2012 May 16. PMID: 22615926; PMCID: PMC3353901.

Li T, Liu B, Chen K, Lou Y, Jiang Y, Zhang D. Small molecule compounds promote the proliferation of chondrocytes and chondrogenic differentiation of stem cells in cartilage tissue engineering. Biomed Pharmacother. 2020 Nov;131:110652. doi: 10.1016/j.biopha.2020.110652. Epub 2020 Sep 14. PMID: 32942151.

Choi SM, Lee KM, Ryu SB, Park YJ, Hwang YG, Baek D, Choi Y, Park KH, Park KD, Lee JW. Enhanced articular cartilage regeneration with SIRT1-activated MSCs using gelatin-based hydrogel. Cell Death Dis. 2018 Aug 29;9(9):866. doi: 10.1038/s41419-018-0914-1. PMID: 30158625; PMCID: PMC6115405.

Author Response

We are sincerely grateful for your thorough review of our manuscript and your valuable comments. We hope that our additions and corrections will enhance the quality of the manuscript text.

INTRODUCTION

Line 68: Enter the abbreviation for lithium chloride (LiCl)

Thank you for your note. The full name has been added (page 2, line 60)

MATERIALS AND METHODS

1 Standardize so that all reagents are standardized with brand and origin. Several of them throughout the text do not have this information.

We indicated the brand and origin of the reagents used in the Methods section (lines 89-93, 95, 110, 117-119, 122, 135-137, 194-195, 218)

2 In the acute knee injury model, it was unclear how many animals were used in total nor how many were used in the group that finished in three weeks and in the group that finished in three months. It was also not detailed how the tissue was removed for immunofluorescence analysis and histology staining, as well as the euthanasia of the animals.

Thank you for noticing this omission. We have added the number of samples studied in the groups for in vitro experiments and the number of animals in the groups for in vivo experiments in the captions of the figures and in the Methods section (lines 167, 212).

3 As shown in the diagram in Figure 1Q, the groups should be presented in the methodology to facilitate understanding of the experiments. A schematic drawing of the study or a graphic abstract at the beginning of the method would be very enlightening.

A description of the amount of samples per group has been provided in the figure captions and in the text of the Methods (page 5, lines 177-181).

4 Was only one concentration of resveratrol used? Testing different concentrations of resveratrol could be more informative. Including a control group trying the other supplements in a micromass culture without using ad matrices could also exclude interference from matrix compounds.

We are grateful for your thoughtful comment. Our thoughts on this matter are as follows.

According to the literature, during the chondrogenic differentiation stage of cell culture to achieve a positive effect on chondrocytes, it is recommended to use 10 and 20 uM of resveratrol (Keshavarz et al., 2020). For WJ-MSCs culture, 17 uM of resveratrol is recommended (Sultan et al., 2020). However, the application of resveratrol during the cell expansion stage requires significantly lower concentrations. We based our approach on the work of Choi et al. (2018), where the authors applied 1 uM of resveratrol and observed a significant increase in cell proliferation, particularly noticeable at the 5th passage. Additionally, Peltz et al. (2012) demonstrated that 1 uM of resveratrol can stimulate MSC proliferation (Peltz et al., 2012) with effects on self-renewal and regeneration starting at a concentration of 0.1 uM, while concentrations above 5 uM of resveratrol, on the contrary, increase the rate of aging.

We consider it reasonable to hypothesize that cell cultures grown without matrices can be used as a control for assessing the influence of the polymer on the course of chondrogenic differentiation. Firstly, matrices of the thermo-sensitive polymer P(NIPAM-co-NtBA) were pre-tested by us, showing no impact on linear cell cultures (Nash et al. 2013; Kazakova et al., 2023), demonstrating no cytotoxicity, and not altering the immunophenotype of cells. This information has been added to the methods section (page 3, lines 133-134). Secondly, it is worth noting that different forms of 3D cultivation influence the growth and differentiation processes of cultures, where cells may respond differently to external stimuli.

For example, ROCK stimulation enhances chondrogenic differentiation in pellets, inhibits in microribbons, but has no effect in hydrogels (Gegg et al., 2020). Resveratrol acts on SIRT1, and SIRT1 inhibits ROCK. Consequently, the cellular response to resveratrol in cell monolayer culture and in micromass culture is expected to differ.

This text has been included into the Discussion (page 10 lines 445-451).

Gegg, C.; Yang, F. The Effects of ROCK Inhibition on Mesenchymal Stem Cell Chondrogenesis Are Culture Model Dependent. Tissue Eng. - Part A 2020, 26, 130–139

Choi et al. Enhanced articular cartilage regeneration with SIRT1-activated MSCs using gelatin-based hydrogel. Cell Death Dis. 2018 Aug 29;9(9):866

Kazakova G.K. et al., Preparation and Characterization of Thermoresponsive Polymer Scaffolds Based on Poly(N-Isopropylacrylamide-Co-N-Tert-Butylacrylamide) for Cell Culture. Technologies 2023, 11, 145, doi:10.3390/technologies11050145.

Keshavarz G et al. Resveratrol Effect on Adipose-Derived Stem Cells Differentiation to Chondrocyte in Three-Dimensional Culture. Adv Pharm Bull. 2020 Jan;10(1):88-96

Nash M.E. et al,. Thermoresponsive Substrates Used for the Expansion of Human Mesenchymal Stem Cells and the Preservation of Immunophenotype. Stem Cell Rev. Reports 2013, 9, 148–157, doi:10.1007/s12015-013-9428-5.

Peltz L et al. Resveratrol exerts dosage and duration dependent effect on human mesenchymal stem cell development. PLoS One. 2012;7(5):e37162

Sultan, S. et al. Resveratrol Promotes Chondrogenesis of Human Wharton’s Jelly Stem Cells in a Hyperglycemic State by Modulating the Expression of Inflammation-Related Cytokines. All Life 2020, 13, 577–586

5 Carrying out clinical assessments regarding the animals' pain, behavior, and walking before and after surgery could provide additional data for the evaluations.

We find this commentary to be extremely valuable. In the experiment, immunodeficient mice were utilized. Surgical procedures were performed in a laminar flow hood within a sterile environment equipped with filters. Behavior and walking tests were not conducted to minimize the time animals spent outside their cages and to eliminate interaction with tools that had come into contact with other laboratory animals, posing serious risks of infection to immunocompromised animals.

The incision of the joint capsule was minimal, and after creating the defect, the patella was returned to its original position and sutured. Following the surgery, animals were kept under observation, and their condition was assessed based on signs of pain and distress, following FELASA guidelines. In the first 24 hours’ post-operation, animals moved only as necessary. Three days after the operation, rats showed no signs of pain. Within one week, the animals moved freely within their cages, and their mobility didn’t differ from that of the intact group.

We have included photographs of the surgical procedure in Supplementary Figure 1 to provide an overview of the size of the inflicted damage. Additionally, we have expanded the description of the animals' condition after the operation in the methods section (page 4, lines 204-210).

RESULTS

1 In Figure 1P, chondrogenic differentiation is not visible. The photo will be presented at a higher magnification to highlight cuboidal cells and gaps around the young chondrocytes.

Thank you for this note, we replaced image on Figure 1P.

2 Figure 1Q should be presented in more detail in the methodology.

The experimental scheme was presented as a separate image in the Methods (Figure 1).

3 Many of the comments in this section could be in the discussion.

Thank you for this comment. We revised the text of the results for discussion examples, but left some comments there, as this allows to understand the intercommection and sequence of experiments

DISCUSSION

In the introduction, the authors mention two successful ones in increasing proliferation and differentiation when they added resveratrol to the chondrogenic differentiation medium. Likewise, they cite successful experiments in chondrogenic differentiation using LiCl and Y2763. Should the authors better explore the studies mentioned and others I suggest below to create hypotheses about why the research cited succeeded in increasing chondrogenesis and why the research presented in this manuscript did not? Could it be due to the concentration of the reactants?

According to the literature, the reagents utilized in our study indeed exhibit dose-dependent effects and can interact differently with various cell populations. During the chondrogenic differentiation stage of cell cultures, to achieve a positive impact on chondrocytes, it is recommended to employ 10 and 20 μM resveratrol (Keshavarz et al., 2020). For WJ-MSCs culture, 17 μM resveratrol is suggested (Sultan et al., 2020).

However, the application of resveratrol during the cell expansion phase requires significantly lower concentrations. We based our approach on the work of Choi et al. (2018) (Choi et al. 2018), where the authors applied 1 μM resveratrol and observed a significant increase in cell proliferation, particularly notable at the 5th passage. Additionally, Peltz L et al. (2012) demonstrated that 1 μM resveratrol can stimulate MSC proliferation (Peltz et al., 2012). The effects of resveratrol on self-renewal and regeneration commence at concentrations of 0.1 μM, whereas concentrations exceeding 5 μM conversely accelerate the aging process.

LiCl and Y27632 also exhibit dose-dependent effects. For instance, the enhancement of MSC aggregation varies with Y27632 dosage between 1 and 50 μM. Additionally, dose-dependent increases in MSC proliferation have been described at LiCl concentrations below 5 μM (Zhu et al., 2020).

In order to prevent unexpected adverse effects during cell sheet culturing with different additive dosages, we relied on reagent concentrations that had previously been reported to have significant chondrogenic effects: 1 μM resveratrol (Choi et al. 2018), 5 μM LiCl (Tanthaisong et al., 2017), and 10 μM Y27632 (Wang, K.-C. et al, 2018).

Choi et al. Enhanced articular cartilage regeneration with SIRT1-activated MSCs using gelatin-based hydrogel. Cell Death Dis. 2018 Aug 29;9(9):866

Keshavarz G et al. Resveratrol Effect on Adipose-Derived Stem Cells Differentiation to Chondrocyte in Three-Dimensional Culture. Adv Pharm Bull. 2020 Jan;10(1):88-96

Peltz L et al. Resveratrol exerts dosage and duration dependent effect on human mesenchymal stem cell development. PLoS One. 2012;7(5):e37162

Sultan, S. et al. Resveratrol Promotes Chondrogenesis of Human Wharton’s Jelly Stem Cells in a Hyperglycemic State by Modulating the Expression of Inflammation-Related Cytokines. All Life 2020, 13, 577–586

Tanthaisong, P., Imsoonthornruksa, S., Ngernsoungnern, A., Ngernsoungnern, P., Ketudat-Cairns, M., & Parnpai, R. (2017). Enhanced chondrogenic differentiation of human umbilical cord wharton’s jelly derived mesenchymal stem cells by GSK-3 Inhibitors. PLoS ONE, 12(1).

Wang, K.-C. et al. (2018). ROCK Inhibition Promotes the Development of Chondrogenic Tissue by Improved Mass Transport. Tissue Engineering Part A, 24(15–16), 1218–1227

Zhu, Z.; Yin, J.; Guan, J.; Hu, B.; Niu, X.; Jin, D.; Wang, Y.; Zhang, C. Lithium Stimulates Human Bone Marrow Derived Mesenchymal Stem Cell Proliferation through GSK-3β-Dependent β-Catenin/Wnt Pathway Activation. FEBS J. 2014, 281, 5371–5389

Or by the solvent used to dissolve the DMSO?

Resveratrol shows low water solubility (<0.05 mg/mL), and according to the manufacturer's recommendation, it is dissolved in dimethyl sulfoxide (DMSO) prior to application. Similarly, the solubility of the reagent Y27632 is higher in DMSO (160 mg/mL) compared to water (90 mg/mL). In numerous studies, DMSO is frequently employed as a solvent, with only a few publications mentioning its non-neutrality and potential influence on cultivation.

However, according to the results of our study, DMSO may not always be a suitable solvent, as it demonstrates its own effects. Information regarding the low water solubility of the reagents has been included in the Methods section (Page 4, lines 148-151).

As well, the Methods section now contains information indicating that DMSO was added in the control experiments in a volume equivalent to the volume of the investigated agents (always 1 μL per 1 mL of medium) (Page 4, lines 154).

Or by the different sources of MSC? Or were the evaluations of other works, not the most appropriate or not as accurate?

We agree with the notion that the source of mesenchymal stem cells (MSCs) and the method of their isolation exert a crucial influence on the ultimate outcome of the experiment. Much of this influence relates to the heterogeneity within the MSC population, determined by both the cell source and the isolation protocol (Costa et al., 2021). Within the isolated MSC population, diverse cell types, including those already undergoing differentiation, are invariably present (Dunn et al., 2021; Wang et al., 2021)

In the pursuit of creating tissue-engineered constructs for the restoration of articular cartilage, MSCs sourced from different origins are employed, utilizing various cultivation technologies. Despite similarities in approaches to chondrogenic differentiation of MSCs, the final outcomes for MSCs derived from distinct sources may exhibit substantial variations (Kurenkova et al., 2022).

It is noteworthy that MSCs from diverse sources possess distinct transcriptomic and proteomic profiles and exhibit disparate preferences for differentiation into specific cellular lineages. Moreover, the individual variability of patients, including their age, gender, and overall health status, should not be overlooked.

These considerations have been integrated into the Discussion section (Page 10, lines 470-480) and further supplemented the Conclusion.

Costa, L.A.; Eiro, N.; Fraile, M.; Gonzalez, L.O.; Saá, J.; Garcia-Portabella, P.; Vega, B.; Schneider, J.; Vizoso, F.J. Functional Heterogeneity of Mesenchymal Stem Cells from Natural Niches to Culture Conditions: Implications for Further Clinical Uses. Cell. Mol. Life Sci. 2021, 78, 447–467, doi:10.1007/s00018-020-03600-0.

Dunn, C.M.; Kameishi, S.; Grainger, D.W.; Okano, T. Strategies to Address Mesenchymal Stem/Stromal Cell Heterogeneity in Immunomodulatory Profiles to Improve Cell-Based Therapies. Acta Biomater. 2021, 133, 114–125, doi:10.1016/j.actbio.2021.03.069.

Kurenkova, A.D.; Romanova, I.A.; Kibirskiy, P.D.; Timashev, P.; Medvedeva, E. V. Strategies to Convert Cells into Hyaline Cartilage: Magic Spells for Adult Stem Cells. Int. J. Mol. Sci. 2022, 23, doi:10.3390/ijms231911169.

Wang, Z.; Chai, C.; Wang, R.; Feng, Y.; Huang, L.; Zhang, Y.; Xiao, X.; Yang, S.; Zhang, Y.; Zhang, X. Single‐cell Transcriptome Atlas of Human Mesenchymal Stem Cells Exploring Cellular Heterogeneity. Clin. Transl. Med. 2021, 11, doi:10.1002/ctm2.650.

Or by the evaluation time?

Indeed, the duration of cultivation during chondrogenic differentiation of MSCs is a significant influencing factor on the final outcome. It has been demonstrated that in WJ-MSC cultures. By the 8th day, there was a predominance of collagen type 1 and 3 fibrotic tissue markers. However, by the 16th day post-differentiation induction, both WJ-MSCs and BM-MSCs in pellet cultures expressed chondrogenic markers such as sox9, collagen II, and aggrecan (Lamparelli et al., 2022). For instance, extending the pellet culture from 3 weeks to 5 weeks intensifies the chondrogenic impact of LiCl, as evidenced by a reduction in the expression of cell hypertrophy markers (Tanthaisong et al. 2017).

Considering this, we extended the cultivation period in the differentiation medium of cell sheets from the conventional 3 weeks to 5 weeks (35 days). It was anticipated that this prolonged duration of chondrogenic differentiation would be sufficient to manifest the chondrogenic effects within the cultures. We deemed it impractical to extend the differentiation of cell sheets beyond 5 weeks.

Lamparelli, E.P.; Ciardulli, M.C.; Giudice, V.; Scala, P.; Vitolo, R.; Dale, T.P.; Selleri, C.; Forsyth, N.R.; Maffulli, N.; Della Porta, G. 3D In-Vitro Cultures of Human Bone Marrow and Wharton’s Jelly Derived Mesenchymal Stromal Cells Show High Chondrogenic Potential. Front. Bioeng. Biotechnol. 2022, 10, 1–18, doi:10.3389/fbioe.2022.986310.

Tanthaisong P, et al. Enhanced chondrogenic differentiation of human umbilical cord Wharton's jelly-derived mesenchymal stem cells by GSK-3 Inhibitors. PLoS One 2017;12(1)

Or could the combination of so many variables have interfered with the final result?

We are in agreement with your assumption. In spite of the published evidence that shows the agents we use (Resveratrol, Y2763 and LiCl) have chondrogenic effects, their operating mechanisms are complex and little understood. This is one of the reasons why we separated resveratrol and chondrogenic agents at different stages of the experiment, and resveratrol was not used at the same time as Y2763 or LiCl during the differentiation stage.

The chondrogenic medium was supplemented with the ROCK and non-canonical Wnt signaling pathway inhibitor Y27632 (Wang et al., 2020) or the canonical Wnt signaling pathway activator LiCl (Tanthaisong et al., 2017), which, according to the literature, can enhance chondrogenic differentiation of MSCs. According to published studies, resveratrol improves MSC proliferation and further chondrogenic differentiation acting through SIRT1 which is ROCK inhibitor (Peltz et al., 2012). However, at the same time effects of ROCK inhibition on MSC chondrogenesis differ substantially depending on culture models (Gegg et al., 2020).

ROCK inhibition using a small molecule of Y2763 led to more efficient retention of the chondrocyte phenotype in human chondrocyte pellet cultures (Matsumoto et al., 2012) and increased chondrogenesis in human MSCs in pellet culture upregulating of chondrogenic signaling (Gegg et al., 2020).

LiCl, through inhibition of GSK3, inactivates the phosphorylation of β-catenin, initiating the canonical Wnt/β-catenin signaling pathway. LiCl should act synergistically with TGF-β3 on chondrogenic differentiation in hWJ-MSC cultures (Tanthaisong et al., 2017). At the same time, the Wnt/β-catenin signaling pathway often crosstalks with other signaling pathways, which interact with each other to modulate chondrogenesis (Hartmann et al., 2016).

As can be seen, the interactions between chondrogenic agents during MSC culturing is complex, multi-component, and frequently overlaps. They have not been sufficiently studied to allow for a clear understanding of the resulting effects.

Discussions were added to the text (Page 10, lines 444-461, lines 470-480).

Gegg, C.; Yang, F. The Effects of ROCK Inhibition on Mesenchymal Stem Cell Chondrogenesis Are Culture Model Dependent. Tissue Eng. - Part A 2020, 26, 130–139

Hartmann, C. Wnt Signaling in Cartilage Development. Cartil. Vol. 1 Physiol. Dev. 2016, 229–252.

Matsumoto, E.; Furumatsu, T.; Kanazawa, T.; Tamura, M.; Ozaki, T. ROCK Inhibitor Prevents the Dedifferentiation of Human Articular Chondrocytes. Biochem. Biophys. Res. Commun. 2012, 420, 124–129,

Peltz L et al. Resveratrol exerts dosage and duration dependent effect on human mesenchymal stem cell development. PLoS One. 2012;7(5):e37162

Tanthaisong, P.; Imsoonthornruksa, S.; Ngernsoungnern, A.; Ngernsoungnern, P.; Ketudat-Cairns, M.; Parnpai, R. Enhanced Chondrogenic Differentiation of Human Umbilical Cord Wharton’s Jelly Derived Mesenchymal Stem Cells by GSK-3 Inhibitors. PLoS One 2017, 12

Wang, K.-C.; Egelhoff, T.T.; Caplan, A.I.; Welter, J.F.; Baskaran, H. ROCK Inhibition Promotes the Development of Chondrogenic Tissue by Improved Mass Transport. Tissue Eng. Part A 2018, 24, 1218–1227, doi:10.1089/ten.tea.2017.0438.

Could a control without using the matrix, such as micromass differentiation, have helped?

We find it reasonable to propose that cell cultures grown without matrices can serve as controls to assess the impact of the polymer on the course of chondrogenic differentiation. However, several considerations merit attention.

Firstly, matrices made of the thermosensitive polymer P(NIPAM-co-NtBA) were pre-tested, and we observed no influence on linear cell cultures. The matrices exhibited no cytotoxicity and did not alter the cell immunophenotype (Nash et al. 2013; Kazakova et al., 2023). This information has been incorporated into the Methods section (page 3, lines 133-134).

Secondly, it is essential to note that the 3D cultivation methods themselves can influence the growth and differentiation processes of mesenchymal stem cells (MSCs), as cells may respond differently to external stimulation. Therefore, cultivating without matrices in the form of micromasses may not serve as a valid control in studying the cultivation of cell pellets.

As an example, MSCs from horse bone marrow cultivated in various types of 3D cultures (pellets, fibrin alginate scaffolds, alginate scaffolds, and agarose scaffolds) exhibited varying levels of expression of chondrogenic and fibrotic tissue markers (Watts et al., 2013).

For instance, stimulation with ROCK enhances chondrogenic differentiation in pellets, inhibits it in microribbons, but has no effect in hydrogels (Gegg et al., 2020). Resveratrol acts on SIRT1, and SIRT1 inhibits ROCK. Consequently, the cellular response to resveratrol in cell pellet culture and micromass culture is expected to differ.

Discussion on this topic has been added to the text (Page 10, lines 448-460).

Gegg, C.; Yang, F. The Effects of ROCK Inhibition on Mesenchymal Stem Cell Chondrogenesis Are Culture Model Dependent. Tissue Eng. - Part A 2020, 26, 130–139

Kazakova G.K. et al., Preparation and Characterization of Thermoresponsive Polymer Scaffolds Based on Poly(N-Isopropylacrylamide-Co-N-Tert-Butylacrylamide) for Cell Culture. Technologies 2023, 11, 145, doi:10.3390/technologies11050145.

Nash M.E. et al,. Thermoresponsive Substrates Used for the Expansion of Human Mesenchymal Stem Cells and the Preservation of Immunophenotype. Stem Cell Rev. Reports 2013, 9, 148–157, doi:10.1007/s12015-013-9428-5.

Watts, A.E.; Ackerman-Yost, J.C.; Nixon, A.J. A Comparison of Three-Dimensional Culture Systems to Evaluate in Vitro Chondrogenesis of Equine Bone Marrow-Derived Mesenchymal Stem Cells. Tissue Eng. - Part A 2013, 19, 2275–2283, doi:10.1089/ten.tea.2012.0479.

Finally, if better discussed, there are many points that could add more value to the manuscript.

We are thankful for all the valuable comments you have provided. We have tried to take them all into account. The following articles have been added to the text of Discussion:

Keshavarz G, Jalili C, Pazhouhi M, Khazaei M. Resveratrol Effect on Adipose-Derived Stem Cells Differentiation to Chondrocyte in Three-Dimensional Culture. Adv Pharm Bull. 2020 Jan;10(1):88-96. doi: 10.15171/apb.2020.011. Epub 2019 Dec 11. PMID: 32002366; PMCID: PMC6983992.

Peltz L, Gomez J, Marquez M, Alencastro F, Atashpanjeh N, Quang T, Bach T, Zhao Y. Resveratrol exerts dosage and duration dependent effect on human mesenchymal stem cell development. PLoS One. 2012;7(5):e37162. doi: 10.1371/journal.pone.0037162. Epub 2012 May 16. PMID: 22615926; PMCID: PMC3353901.

Li T, Liu B, Chen K, Lou Y, Jiang Y, Zhang D. Small molecule compounds promote the proliferation of chondrocytes and chondrogenic differentiation of stem cells in cartilage tissue engineering. Biomed Pharmacother. 2020 Nov;131:110652. doi: 10.1016/j.biopha.2020.110652. Epub 2020 Sep 14. PMID: 32942151.

Choi SM, Lee KM, Ryu SB, Park YJ, Hwang YG, Baek D, Choi Y, Park KH, Park KD, Lee JW. Enhanced articular cartilage regeneration with SIRT1-activated MSCs using gelatin-based hydrogel. Cell Death Dis. 2018 Aug 29;9(9):866. doi: 10.1038/s41419-018-0914-1. PMID: 30158625; PMCID: PMC6115405.